

# Universal scaling of quench-induced correlations in a one-dimensional channel at finite temperature

Alessio Calzona[1,2,3,*], Filippo M. Gambetta[1,2], Matteo Carrega[4], Fabio Cavaliere[1,2], Thomas L. Schmidt[3] and Maura Sassetti[1,2]

**1** Dipartimento di Fisica, Università di Genova, Via Dodecaneso 33, I-16146, Genova, Italy
**2** SPIN-CNR, Via Dodecaneso 33, I-16146, Genova, Italy
**3** Physics and Materials Science Research Unit, University of Luxembourg, 162a avenue de la Faïencerie, L-1511 Luxembourg
**4** NEST, Istituto Nanoscienze-CNR and Scuola Normale Superiore, Piazza San Silvestro 12, I-56127 Pisa, Italy

* calzona@fisica.unige.it

## Abstract

It has been shown that a quantum quench of interactions in a one-dimensional fermion system at zero temperature induces a universal power law $\propto t^{-2}$ in its long-time dynamics. In this paper we demonstrate that this behaviour is robust even in the presence of thermal effects. The system is initially prepared in a thermal state, then at a given time the bath is disconnected and the interaction strength is suddenly quenched. The corresponding effects on the long times dynamics of the non-equilibrium fermionic spectral function are considered. We show that the non-universal power laws, present at zero temperature, acquire an exponential decay due to thermal effects and are washed out at long times, while the universal behaviour $\propto t^{-2}$ is always present. To verify our findings, we argue that these features are also visible in transport properties at finite temperature. The long-time dynamics of the current injected from a biased probe exhibits the same universal power law relaxation, in sharp contrast with the non-quenched case which features a fast exponential decay of the current towards its steady value, and thus represents a fingerprint of quench-induced dynamics. Finally, we show that a proper tuning of the probe temperature, compared to that of the one-dimensional channel, can enhance the visibility of the universal power-law behaviour.

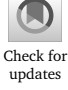

## 1  Introduction

Among all open problems in the field of quantum many-body systems, of special interest is the study of real-time dynamics far from equilibrium. In this context, recent state-of-the-art experiments performed on cold atoms [1–7] have shown the possibility to modulate in time various parameters and to detect transport properties as a useful tool to investigate the system response [8–11]. Due to their high degree of tunability, cold atoms represent an ideal platform to study *quantum quenches* [12–14], which consist in the rapid variation over time of one of the system parameters in a controlled way. Such protocol naturally settles an out-of-equilibrium state with highly non-trivial time evolution. Non-equilibrium physics of one-dimensional (1D) systems and time-resolved dynamics has been also recently investigated in pioneering experiments in solid-state implementation, see for example [15–19].

When dealing with out-of-equilibrium systems, two important questions arise: do such systems eventually settle to a steady state? And if so, what characterises their relaxation dynamics? Several theoretical studies [20–23] and experiments have addressed such topics, providing important results which strongly depend on the system considered. For example, some cold atom systems have shown a faster-than-expected relaxation towards a state well described by a grand-canonical ensemble [5]. Quantum quench experiments performed splitting a 1D gas of cold atoms [4], initially prepared in a thermal state, have shown the system to settle towards a quasi-thermal state, which can be well-described in terms of an effective temperature lower than the initial one. On the contrary, other cold-atom implementations have reported relaxation towards a steady state which cannot be described as a thermal one [2, 24]. In this respect, integrable systems are particularly interesting, since it has been conjectured that they approach a steady state that can be characterised by the so-called generalised Gibbs ensemble [25, 26], whose associated density matrix is in general very different from the thermal one. However, a complete characterisation of their relaxation dynamics is yet to be found.

Among all 1D integrable systems, a special role is played by Luttinger liquids [27, 28] which have been detected in several experiments [4, 6, 7, 29, 30]. Indeed, they are characterised by an infinite number of conserved quantities which allow to promptly construct their generalised Gibbs ensemble. Moreover, the unavoidable presence of interactions give rise to several intriguing effects including charge and spin fractionalisation [16–18, 28, 31–43], which have been also investigated in the presence of a quantum quench [44, 45]. Relaxation dynamics of Luttinger liquids have been the subject of recent theoretical activity [12, 14, 46–53], even at finite temperature [54–56].

In a recent paper [45], the relaxation dynamics of a Luttinger liquid subjected to a sudden quench of the interaction strength has been studied at zero temperature. It has been shown that the latter generates entanglement between counter-propagating excitations [45, 57], resulting in finite bosonic cross-correlators among the 1D channel. This, in turn, affects the time evolution of fermionic observables such as the charge and the energy currents locally injected from a biased probe. Indeed, in the long-time limit they display a universal power-law

relaxation dynamics $\propto t^{-2}$, not sensitive to the details of the quench.

In this paper, we investigate whether this universal power law is still present when the system is prepared, before the quench, in a thermal state. We therefore study the behaviour of the fermionic non-equilibrium spectral function and its relaxation dynamics toward the steady state at finite temperature. We demonstrate that at long times its universal quench decay $\propto t^{-2}$ is stable against thermal effects, while the non-universal interaction-induced power laws, present at $T = 0$, acquire a fast exponential decay dictated by the finite initial temperature. The initial preparation in a thermal state can then be useful to highlight and enhance the visibility of universal features in comparison to the zero temperature case. We also consider the impact of this quench protocol with finite temperature preparation on transport. The behaviour of the spectral function, indeed, determines the relaxation dynamics of the charge current injected from a biased probe, which displays a universal scaling law $\propto t^{-2}$, again robust against thermal effects. This is in sharp contrast with a conventional non-quenched situation, in which a finite temperature induces a fast, non-universal exponential decay towards the steady current value. We also consider the case of different initial temperatures for the 1D channel and the probe, showing that a proper tuning of the latter can enhance significantly the visibility of the $\propto t^{-2}$ universal decay in the current dynamics.

The outline of the paper is the following. In Sec. 2 the model is described. In Sec. 3 the fermionic Green functions, central to the evaluation of the transport properties, are introduced. Their basic building blocks, the bosonic two-point correlation functions are also defined and evaluated, and their time dynamics is analysed in detail. In Sec. 4 we focus on the non-equilibrium spectral function and its relaxation dynamics towards the steady state. In Sec. 5 we consider a possible transport setup, where a probe is assumed to be weakly tunnel-coupled to 1D channel, and we evaluate the resulting charge current and its time-evolution after the quench. Section 6 contains the conclusions.

## 2 Model

We consider an interacting 1D channel of spinless fermions with short-range repulsive interactions. At times $t < 0$, they are governed by the Hamiltonian (in all the paper we set $\hbar = k_B = 1$)

$$H_i = v \sum_{r=R,L} \vartheta_r \int_{-\infty}^{+\infty} dx \, \psi_r^\dagger(x)(-i\partial_x)\psi_r(x) + H_i^{(int)}, \tag{1}$$

with $v$ the Fermi velocity, $\psi_r(x)$ the fermion field of the $r$-channel ($r = L/R$ for left-/right-branches), and $\vartheta_{R/L} = +/-$. The interaction term is given by

$$H_i^{(int)} = \frac{g_i^{(4)}}{2} \sum_r \int_{-\infty}^{+\infty} dx \, [n_r(x)]^2 + g_i^{(2)} \int_{-\infty}^{+\infty} dx \, n_R(x)n_L(x), \tag{2}$$

where $n_r(x) =: \psi_r^\dagger(x)\psi_r(x) :$ is the particle density on the $r$-channel and $g_i^{(2)}$, $g_i^{(4)}$ are the coupling strengths of the intra- and inter-channel interactions, respectively [27,28]. For simplicity, hereafter we will consider the Galilean invariant case $g_i^{(2)} = g_i^{(4)} = g_i$. This fact has no relevant consequence, as the more general case $g_i^{(2)} \neq g_i^{(4)}$ leads to the same qualitative results discussed here.

The non-interacting part of $H_i$ can be written in terms of free bosonic fields $\phi_r(x)$, related to the fermionic field operator by the bosonization identity [27,28]

$$\psi_r(x) = \frac{1}{\sqrt{2\pi a}} e^{-i\sqrt{2\pi}\phi_r(x)} \, e^{i\vartheta_r q_F x}, \tag{3}$$

where $q_F$ is the Fermi wave vector and $a$ is the usual short-length cut-off properly introduced in bosonization techniques [28]. Here, we have safely omitted Klein factors. Different regularisation schemes are possible and have been considered in the literature, see for example the constructive bosonization approach described in [58, 59]. The Hamiltonian in Eq. (1) can then be diagonalised introducing new bosonic operators $\phi_{i,\pm}(x)$, connected to $\phi_r(x)$ by the canonical transformation

$$\phi_r(x) = \sum_{\eta=\pm} B_{\eta\vartheta_r} \phi_{i,\eta}(x), \tag{4}$$

with the Bogoliubov coefficients $2B_\pm = K_i^{-1/2} \pm K_i^{1/2}$ and

$$K_i = \sqrt{\frac{1}{1 + \frac{g_i}{\pi v}}} \tag{5}$$

a parameter encoding the repulsive interaction strength, with $0 < K_i \leq 1$ and $K_i = 1$ for non-interacting fermions. The Hamiltonian $H_i$ then reads

$$H_i = \frac{v_i}{2} \sum_{\eta=\pm} \int_{-\infty}^{+\infty} dx \left[ \partial_x \phi_{i,\eta}(x) \right]^2, \tag{6}$$

where $v_i = v/K_i$ is the *renormalised* propagation velocity of the collective excitations described by the chiral fields

$$\phi_{i,\eta}(x,t) = \phi_{i,\eta}(x - \eta v_i t, 0) \qquad (t < 0). \tag{7}$$

For $t < 0$ the channel is prepared in a thermal state described by the equilibrium density matrix $\rho_{eq} = Z^{-1} e^{-H_i/T}$, where $Z = \text{Tr}\{e^{-H_i/T}\}$ is the partition function.

At $t = 0$, the bath is disconnected and the interaction strength is suddenly quenched $g_i \to g_f$ ($K_i \to K_f$), resulting in a sudden switch of the Hamiltonian $H_i \to H_f$ with

$$H_f = \frac{v_f}{2} \sum_{\eta=\pm} \int_{-\infty}^{+\infty} dx \left[ \partial_x \phi_{f,\eta}(x) \right]^2. \tag{8}$$

Here, $v_f = v/K_f$ and $K_f = \left[ 1 + (g_f/\pi v) \right]^{-1/2}$ describes the interaction strength in the post-quench state. Note that also these fields obey chiral properties,

$$\phi_{f,\eta}(x,t) = \phi_{f,\eta}(x - \eta v_f t, 0) \qquad (t > 0), \tag{9}$$

and they are connected to $\phi_{i,\eta}(x,t)$ by the canonical transformation [44, 51]

$$\phi_{f,\eta}(x,t) = \sum_{\ell=\pm} \theta_{\ell\eta} \phi_{i,\ell}(x,t), \quad \text{with} \quad \theta_\pm = \frac{1}{2} \left[ \sqrt{\frac{1}{\epsilon}} \pm \sqrt{\epsilon} \right], \tag{10}$$

with

$$\epsilon = \frac{K_i}{K_f}, \tag{11}$$

and to the free bosonic fields by

$$\phi_r(x,t) = \sum_{\eta=\pm} A_{\eta\vartheta_r} \phi_{f,\eta}(x,t) \quad \text{with} \quad A_\pm = \frac{1}{2} \left[ \sqrt{\frac{1}{K_f}} \pm \sqrt{K_f} \right]. \tag{12}$$

Note that the non-quenched case ($K_i = K_f$) is represented by $\theta_- = 0$, $\theta_+ = 1$ and $A_\pm = B_\pm$. Finally, the different Bogoliubov coefficients are related among themselves via $B_\pm = A_\pm \theta_+ + A_\mp \theta_-$.

# 3 Fermionic and bosonic correlation functions

In the following we will discuss the local lesser fermionic Green function of the 1D channel

$$G_r^<(z, t, \bar{t}) \equiv i \left\langle \psi_r^\dagger(z, \bar{t}) \psi_r(z, t) \right\rangle_{eq} = \frac{i}{2\pi a} \left\langle e^{i\sqrt{2\pi}\phi_r(z,\bar{t})} e^{-i\sqrt{2\pi}\phi_r(z,t)} \right\rangle_{eq} \tag{13}$$

at the generic position $z$. Here, the brackets $\langle \ldots \rangle_{eq}$ denotes a quantum average performed on the initial thermal density matrix $\rho_{eq}$ and the last bosonic expression is obtained using the identity of Eq. (3). The first step to evaluate the Green function is to compute the time evolution of the free bosonic fields $\phi_r(z, t)$. It crucially depends on whether the operators are evaluated before or after the quench. Indeed, using Eqs. (4) and (12) one can write

$$\phi_r(z, t) = \begin{cases} A_{\vartheta_r} \phi_{f,+}(z - v_f t, 0) + A_{-\vartheta_r} \phi_{f,-}(z + v_f t, 0) & t > 0 \\ B_{\vartheta_r} \phi_{i,+}(z - v_i t, 0) + B_{-\vartheta_r} \phi_{i,-}(z + v_i t, 0) & t < 0 \end{cases}, \tag{14}$$

having exploited the proper chirality properties of Eq. (7) and Eq. (9).

Note that, since space translational invariance is not broken by the quench protocol, the Green function will not depend on the generic position $z$. On the other hand, it will feature four different time regimes, depending on the sign of $t$ and $\bar{t}$, due to the breaking of time translational invariance. One can thus write

$$\begin{aligned} G_r^<(t, \bar{t}) &= \vartheta(-t)\vartheta(-\bar{t}) \mathscr{G}_r^{nn}(t, \bar{t}) + \vartheta(t)\vartheta(-\bar{t}) \mathscr{G}_r^{pn}(t, \bar{t}) \\ &+ \vartheta(-t)\vartheta(\bar{t}) \mathscr{G}_r^{np}(t, \bar{t}) + \vartheta(t)\vartheta(\bar{t}) \mathscr{G}_r^{pp}(t, \bar{t}), \end{aligned} \tag{15}$$

where $\vartheta(t)$ is the unit step function. If both time arguments are negative ($n$), the Green function is not affected by the subsequent quench and it thus reduces to the standard equilibrium result

$$\mathscr{G}_r^{nn}(t, \bar{t}) = \frac{i}{2\pi a} \exp\left[ 2\pi \left( B_+^2 + B_-^2 \right) \mathscr{C}_+(v_i(t - \bar{t})) \right] \qquad (t, \bar{t} < 0). \tag{16}$$

Here, we have introduced the equal time bosonic correlator

$$\mathscr{C}_+(x) \equiv \langle \phi_{i,+}(x, 0)\phi_{i,+}(0, 0) \rangle_{eq} - \langle \phi_{i,+}^2(0, 0) \rangle_{eq}, \tag{17}$$

whose expectation value on the initial thermal state is [28]

$$\mathscr{C}_+(x) = \frac{1}{2\pi} \ln \frac{\left| \Gamma(1 + T\omega_i^{-1} - iTxv_i^{-1}) \right|^2}{\Gamma(1 + T\omega_i^{-1})^2} + \frac{1}{2\pi} \ln\left( \frac{1}{1 - ixa^{-1}} \right), \tag{18}$$

with $\Gamma(z)$ the Euler Gamma function and $\omega_i = v_i a^{-1}$ a cut-off frequency. From now on, we will safely set $a^{-1} = q_F$. It is worth to notice that $\mathscr{G}_r^{nn}(t, \bar{t})$ depends only on the time difference $\tau = t - \bar{t}$. By contrast, if $t > 0$ and $\bar{t} < 0$ one has

$$\begin{aligned} \mathscr{G}_r^{pn}(t, \bar{t}) = \frac{i}{2\pi a} \exp\Big[ &2\pi\, \theta_+ (B_+ A_+ + B_- A_-)\, \mathscr{C}_+(v_i(t - \bar{t}) + (v_f - v_i)t) \\ &+ 2\pi\, \theta_- (B_+ A_- + B_- A_+)\, \mathscr{C}_+(v_i(t - \bar{t}) - (v_f + v_i)t) \\ &+ 2\pi A_+ A_- \theta_+ \theta_- (\mathscr{C}_+(2v_f t) + \mathscr{C}_+(-2v_f t)) \Big]. \end{aligned} \tag{19}$$

The opposite regime ($t < 0$, $\bar{t} > 0$) can be easily obtained by exploiting $\mathscr{G}_r^{np}(t, \bar{t}) = -\mathscr{G}_r^{pn}(\bar{t}, t)^*$.

Finally, the fourth regime, when both $t$ and $\bar{t}$ are positive, is the most interesting one and can be expressed as

$$
\begin{aligned}
\mathscr{G}_r^{pp}(t,\bar{t}) = \frac{i}{2\pi a} \exp\Big[ &2\pi\,\theta_+^2(A_+^2+A_-^2)\,\mathscr{C}_+(-v_f(t-\bar{t})) + 2\pi\,\theta_-^2(A_+^2+A_-^2)\,\mathscr{C}_+(v_f(t-\bar{t})) \\
&+ 2\pi A_-A_+\theta_+\theta_-(\mathscr{C}_+(v_f(t+\bar{t})) + \mathscr{C}_+(-v_f(t+\bar{t}))) \\
&- 2\pi A_-A_+\theta_+\theta_-(\mathscr{C}_+(v_f t) + \mathscr{C}_+(-v_f t) + \mathscr{C}_+(2v_f\bar{t}) + \mathscr{C}_+(-2v_f\bar{t})) \Big].
\end{aligned}
\tag{20}
$$

Indeed, as we will see, it is the only one responsible for the presence of the universal features which characterise the post-quench relaxation dynamics. Moreover, it is the only one controlling the transport properties after the quench. It is therefore worth to analyse it in details, highlighting the physical origin of its peculiar quench-induced features. To this end, it is useful to rewrite it in terms of correlation functions between the final chiral bosonic fields. This reads

$$
\mathscr{G}_r^{pp}(t,\bar{t}) = \frac{i}{2\pi a} \exp\Big\{\pi\Big[ A_{\vartheta_r}^2 D_{+,+}(t,t-\bar{t}) + A_{-\vartheta_r}^2 D_{-,-}(t,t-\bar{t}) + 2A_+A_- D_{+,-}(t,t-\bar{t}) \Big]\Big\}, \tag{21}
$$

with

$$
\begin{aligned}
D_{\alpha,\beta}(t,\tau) \equiv\; &2\langle\phi_{f,\alpha}(0,t-\tau)\phi_{f,\beta}(0,t)\rangle_{eq} - \langle\phi_{f,\alpha}(0,t-\tau)\phi_{f,\beta}(0,t-\tau)\rangle_{eq} \\
&- \langle\phi_{f,\alpha}(0,t)\phi_{f,\beta}(0,t)\rangle_{eq}.
\end{aligned}
\tag{22}
$$

Here $\alpha,\beta = \pm$ and the time difference should satisfy $\tau \equiv t-\bar{t} < t$. Exploiting the chirality of the fields as well as Eqs. (10) and (18), these correlation functions can be decomposed as

$$
D_{\alpha,\beta}(t,\tau) = D_{\alpha,\beta}^{(0)}(t,\tau) + \Delta D_{\alpha,\beta}(t,\tau), \tag{23}
$$

with $D_{\alpha,\beta}^{(0)}(t,\tau)$ the zero-temperature contribution and $\Delta D_{\alpha,\beta}(t,\tau)$ corrections due to the finite-temperature of the initial state. In particular, one obtains

$$
D_{\alpha,\alpha}^{(0)}(t,\tau) = \sum_{\eta=\pm} \frac{\theta_\eta^2}{\pi} \log\frac{1}{1-i\eta\omega_f\tau}, \tag{24}
$$

$$
\Delta D_{\alpha,\alpha}(t,\tau) = \frac{\theta_+^2+\theta_-^2}{\pi} \log\frac{|\Gamma(1+T\omega_i^{-1}+iT\epsilon\tau)|^2}{\Gamma^2(1+T\omega_i^{-1})}, \tag{25}
$$

$$
D_{\alpha,-\alpha}^{(0)}(t,\tau) = \frac{\theta_+\theta_-}{2\pi} \log\frac{[1+4\omega_f^2(t-\tau)^2][1+4\omega_f^2 t^2]}{[1+\omega_f^2(2t-\tau)^2]^2}, \tag{26}
$$

$$
\Delta D_{\alpha,-\alpha}(t,\tau) = \frac{2\theta_+\theta_-}{\pi} \log\frac{|\Gamma(1+T\omega_i^{-1}-iT\epsilon(2t-\tau))|^2}{|\Gamma(1+T\omega_i^{-1}+2iT\epsilon(t-\tau))||\Gamma(1+T\omega_i^{-1}+2iT\epsilon t)|}, \tag{27}
$$

with $\epsilon$ defined in Eq. (11). As a general feature, we note that auto-correlators ($\alpha=\beta$) only depend on $\tau$ and not on $t$. In contrast, cross-correlators ($\alpha=-\beta$) feature a full dependence on both $t$ and $\tau$. A direct comparison between the quenched and the non-quenched cases turns out to be particularly useful. In the latter one has $\theta_- = 0$ and cross-correlators vanish, i.e.

$$
D_{\alpha,-\alpha}^{(0)}(t,\tau) = \Delta D_{\alpha,-\alpha}(t,\tau) = 0 \quad \text{if } K_i = K_f. \tag{28}
$$

This is a consequence of the fact that chiral fields $\phi_{f,\eta}(x,t)$ are completely decoupled from each other in the Hamiltonian $H_f$. Conversely, we observe that a quantum quench always leads to finite cross-correlations, see Eqs. (26) and (27).

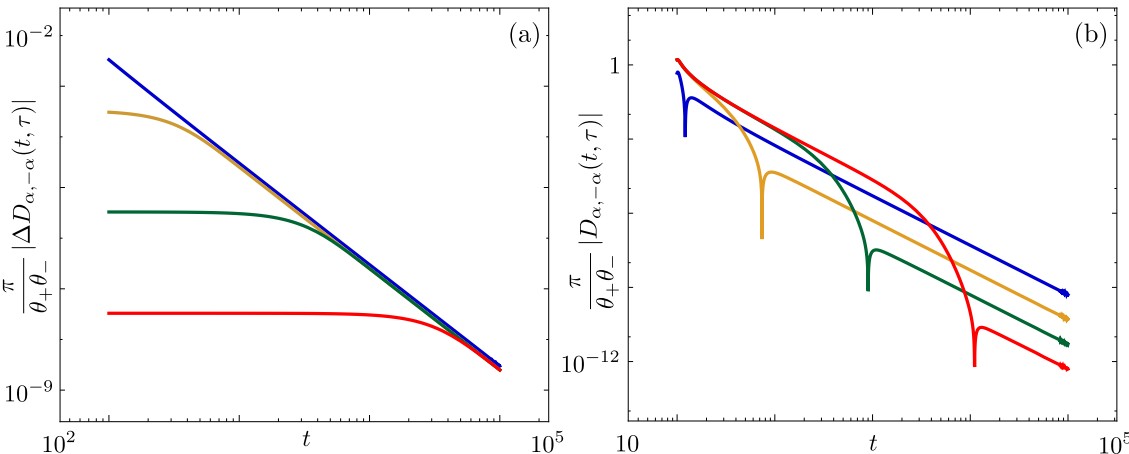

Figure 1: (a) Plot of $\frac{\pi}{\theta_+\theta_-}|\Delta D_{\alpha,-\alpha}(t,\tau)|$ as a function of time $t$ for different temperatures: $T = 10^{-4}$ (red), $T = 10^{-3}$ (green), $T = 10^{-2}$ (yellow), $T = 10^{-1}$ (blue). (b) Plot of the full cross-correlator $\frac{\pi}{\theta_+\theta_-}|D_{\alpha,-\alpha}(t,\tau)|$ as a function of time $t$ for different temperatures: $T = 10^{-4}$ (red), $T = 10^{-3}$ (green), $T = 10^{-2}$ (yellow), $T = 10^{-1}$ (blue). Here, $\tau = 10$, $K_i = 0.9$, $K_f = 0.6$, time units $\omega_f^{-1}$ and temperature units $\omega_i$.

To study the relaxation dynamics it is useful to derive asymptotic expressions for $D_{\alpha,-\alpha}(t,\tau)$. For the zero-temperature contribution of $D^{(0)}_{\alpha,-\alpha}(t,\tau)$ the relevant time scale is $\tau$ and for $t \gg \tau$ one finds via [45]

$$D^{(0)}_{\alpha,-\alpha}(t,\tau) = -\frac{\theta_+\theta_-}{\pi}\left(\frac{\tau}{2t}\right)^2 + O\left(\frac{\tau}{t}\right)^3. \tag{29}$$

Inspecting the temperature-dependent term $\Delta D_{\alpha,-\alpha}(t,\tau)$, an additional time scale $(\epsilon T)^{-1}$ emerges and two different regimes of the gamma functions, present in Eq. (27), have to be considered

$$\left|\Gamma(1 + T\omega_i^{-1} + iT\epsilon t)\right|^2 \approx \begin{cases} 2\pi(T\epsilon t)^{1+2T\omega_i^{-1}}e^{-\pi T\epsilon t} & t \gg (\epsilon T)^{-1} \\ \Gamma^2(1 + T\omega_i^{-1})\left[1 - (T\epsilon t)^2\,\Gamma(1, 1 + T\omega_i^{-1})\right] & t \ll (\epsilon T)^{-1} \end{cases}, \tag{30}$$

where $\Gamma(n, z)$ is the $n$-th order polygamma function. Inserting Eq. (30) in the expression for $\Delta D_{\alpha,-\alpha}(t,\tau)$ of Eq. (27) and considering the reasonable temperature regime $T \ll \omega_i$, one obtains the leading terms

$$\Delta D_{\alpha,-\alpha}(t,\tau) \approx \frac{\theta_+\theta_-}{\pi}\begin{cases} 2(T\epsilon\tau)^2\,\Gamma(1, 1 + T\omega_i^{-1}) & \tau \ll t \ll (\epsilon T)^{-1} \\ (1 + 2T\omega_i^{-1})\left(\frac{\tau}{2t}\right)^2 & \tau \ll (\epsilon T)^{-1} \ll t \end{cases}. \tag{31}$$

The validity of this asymptotic expansion is confirmed in Fig. 1(a), where we have numerically evaluated $\Delta D_{\alpha,-\alpha}(t,\tau)$ as a function of time $t$ for different temperatures and at fixed $\tau$. At short time $t \ll (\epsilon T)^{-1}$ the thermal component of the cross-correlator saturates to a time-independent value, while a power law decay $\propto t^{-2}$ is evident for $t \gg (\epsilon T)^{-1}$, in accordance with Eq. (31).

Combining Eqs. (29) and (31) we obtain the asymptotic expression for the full cross-correlator

$$D_{\alpha,-\alpha}(t,\tau) \approx \frac{\theta_+\theta_-}{\pi}\begin{cases} 2(T\epsilon\tau)^2\,\Gamma(1, 1 + T\omega_i^{-1}) - \left(\frac{\tau}{2t}\right)^2 & \tau \ll t \ll (\epsilon T)^{-1} \\ 2T\omega_i^{-1}\left(\frac{\tau}{2t}\right)^2 & \tau \ll (\epsilon T)^{-1} \ll t \end{cases}. \tag{32}$$

As one can see there are two different regimes, both featuring a power-law decay $\propto t^{-2}$ but with different prefactors. For long time $t \gg (\epsilon T)^{-1}$ the cross-correlator is positive and proportional to the temperature. By contrast, when finite temperature effects have not yet kicked in, i.e. at shorter time $t \ll (\epsilon T)^{-1}$, the prefactor of the $t^{-2}$ power-law decay has a negative prefactor and is temperature-independent. This behaviour emerges clearly in Fig. 1(b) where we plotted $|D_{\alpha,-\alpha}(t,\tau)|$ for fixed $\tau$. The transition between the two regimes spans an order of magnitude.

## 4 Non-equilibrium spectral function

To fully characterise the effects of the quench, we now focus on the local (lesser) non-equilibrium spectral function. This is a key quantity to inspect the presence of universal features in the relaxation dynamics and where the role of finite temperature plays a non-trivial role. Moreover, it is a important ingredient to evaluate transport properties. The local non-equilibrium spectral function is defined as [48]

$$A_r^<(\omega,t) \equiv \frac{-i}{2\pi} \int_{-\infty}^{\infty} e^{i\omega\tau} G_r^<(t,t-\tau)\, d\tau. \tag{33}$$

Our task is to investigate its time evolution after the quench, i.e. for $t > 0$. Since the integration range over $\tau$ extends to $+\infty$, the calculation of the above expression requires to distinguish between two different regimes of the Green function:

$$\begin{aligned}
A_r^<(\omega,t) &= \mathscr{A}_r^{pp}(\omega,t) + \mathscr{A}_r^{pn}(\omega,t) \\
&= \frac{-i}{2\pi} \int_{-\infty}^{t} e^{i\omega\tau} \mathscr{G}_r^{pp}(t,t-\tau)\, d\tau + \frac{-i}{2\pi} \int_{t}^{\infty} e^{i\omega\tau} \mathscr{G}_r^{pn}(t,t-\tau)\, d\tau.
\end{aligned} \tag{34}$$

Equation (34) is exact and suitable for numerical investigation, using Eqs. (19,20).

This quantity admits a finite steady state $\bar{A}_r(\omega) \equiv \lim_{t\to\infty} A_r^<(\omega,t)$ given by

$$\begin{aligned}
\bar{A}_r(\omega) &= \frac{-i}{2\pi} \int_{-\infty}^{+\infty} e^{i\omega\tau} \lim_{t\to\infty} \mathscr{G}_r^{pp}(t,t-\tau)\, d\tau \\
&= \frac{a^{-1}}{(2\pi)^2} \int_{-\infty}^{\infty} e^{i\omega\tau} \left( \frac{1}{1-i\omega_f\tau} \right)^{\nu_-} \left( \frac{1}{1+i\omega_f\tau} \right)^{\nu_+} \left[ \frac{|\Gamma(1+T\omega_i^{-1}+iT\epsilon\tau)|^2}{\Gamma^2(1+T\omega_i^{-1})} \right]^{\nu_+ + \nu_-} \mathrm{d}\tau,
\end{aligned} \tag{35}$$

with

$$\nu_\pm = \theta_\mp^2 (A_+^2 + A_-^2). \tag{36}$$

It is instructive to derive some helpful analytical expansions. In particular, we can observe that the integrand $e^{i\omega\tau} \mathscr{G}_r^{pp}(t,t-\tau)$ contributes to $\mathscr{A}_r^{pp}(\omega,t)$ only in two distinct regions: near $\tau \sim 0$, where the Green function approaches its poles - see Eqs.(20,24) - and close to the boundary of the integration domain $\tau \sim t$. As a consequence, in the long-time limit $t \gg \omega^{-1}$ one has [45]

$$\mathscr{A}_r^{pp}(\omega,t) \approx \mathscr{A}_r^{(1)}(\omega,t) + \mathscr{A}_r^{(2)}(\omega,t), \tag{37}$$

the former term stemming from an expansion of the integrand for $\tau \to 0$ and the latter from an expansion for $\tau \to t$. In particular, using Eq. (31), one finds

$$\mathscr{A}_r^{(1)}(\omega,t) = \bar{A}_r(\omega) - \kappa(t)\gamma \frac{\partial_\omega^2 \bar{A}_r(\omega)}{t^2} + O\left(\frac{1}{t}\right)^3, \tag{38}$$

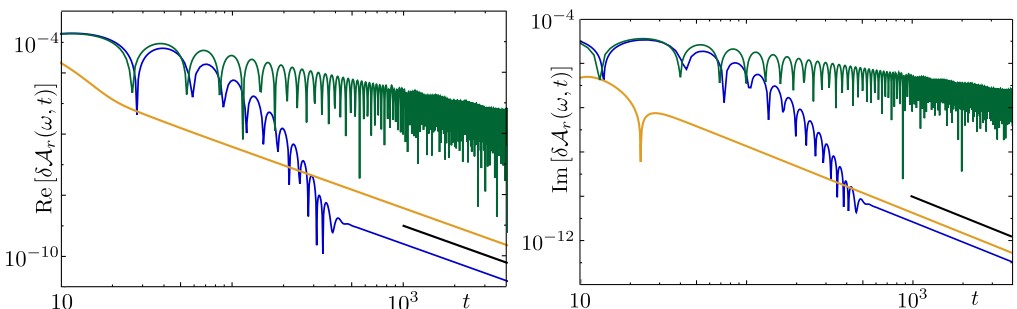

Figure 2: Plot of real (left panel) and imaginary (right panel) part of $\delta A_r(\omega, t) = A_r^<(\omega, t) - \bar{A}_r(\omega)$ (units $v_f^{-1}$) as a function of time for different temperatures: $T = 10^{-1}$ (orange), $T = 10^{-2}$ (blue), $T = 0$ (green). Black lines show a power-law decay $\propto t^2$ (left panel) and $\propto t^{-3}$ (right panel). Here $\omega = 0.1\omega_f$, $K_i = 0.9$, $K_f = 0.6$, time units $\omega_f^{-1}$ and temperature units $\omega_i$. We have included the zero temperature case (green curve) to underline the difference with the finite temperature case.

where

$$\kappa(t) = \begin{cases} -\frac{1}{2} & t \ll (\epsilon T)^{-1} \\ T\omega_i^{-1} & t \gg (\epsilon T)^{-1} \end{cases} \qquad \text{and} \qquad \gamma = -A_+ A_- \theta_+ \theta_-. \tag{39}$$

Considering $\mathscr{A}_r^{(2)}(\omega, t)$, it is useful to introduce $z = t - \tau$ and expand the integrand retaining only the leading term in $z/t$. The result reads

$$\mathscr{A}_r^{(2)}(\omega, t) = \frac{e^{i\omega t}}{a} \frac{\left| \Gamma(1 + T\omega_i^{-1} + 2iT\epsilon t) \right|^{4\gamma} \left| \Gamma(1 + T\omega_i^{-1} + iT\epsilon t) \right|^{2(\nu_+ + \nu_- - 4\gamma)}}{(\omega_f t)^{\nu_+ + \nu_- - 2\gamma}} f(T, \omega), \tag{40}$$

with

$$f(T, \omega) = \frac{1}{(2\pi)^2 4^\gamma} \left[ \Gamma(1 + T\omega_i^{-1}) \right]^{-2(\nu_+ + \nu_-)} \int_0^\infty dz \, e^{-i\omega z} \left[ \frac{\left| \Gamma(1 + T\omega_i^{-1} + 2iT\epsilon z) \right|^4}{1 + 4\omega_f^2 z^2} \right]^\gamma. \tag{41}$$

Using Eq. (30), two asymptotic regimes can be identified

$$\mathscr{A}_r^{(2)}(\omega, t) \approx f(T, \omega) \frac{e^{i\omega t}}{a} \begin{cases} \left[ \Gamma^2(1 + T\omega_i^{-1}) \right]^\xi (\omega_f t)^{-\xi} & t \ll (\epsilon T)^{-1} \\ 4^\gamma (2\pi)^{2\xi} \left[ \frac{(T\epsilon t)^{1 + 2T\omega_i^{-1}}}{\omega_f t} \right]^\xi \exp\left[ -2\pi T\epsilon t(\nu_+ + \nu_-) \right] & t \gg (\epsilon T)^{-1} \end{cases}, \tag{42}$$

with

$$\xi = \nu_+ + \nu_- - 2\gamma. \tag{43}$$

a non-universal exponent encoding the strength of the interaction quench which, for reasonable values of the interaction quench, can take values [45] $1 < \xi \leq 2$. The first regime, which is present as long as $t \ll (\epsilon T)^{-1}$, is a typical interaction-dependent power law. On the other hand, for $t \gg (\epsilon T)^{-1}$, a fast exponential decay appears which basically kills $\mathscr{A}_r^{(2)}(\omega, t)$ in the long-time limit.

The same arguments can be used to derive the asymptotic behaviour of $\mathscr{A}_r^{pn}(\omega, t)$ present in Eq. (34). In this case, the integrand contributes only for $\tau \sim t$ and $\tau \sim t(\nu_i + \nu_f)\nu_i^{-1}$. The final expression reads

$$\mathscr{A}_r^{pn}(\omega, t) = \mathscr{A}_r^{(3)}(\omega, t) + \mathscr{A}_r^{(4)}(\omega, t), \tag{44}$$

where the contribution stemming from the expansion for $\tau \to t$ is

$$
\mathscr{A}_r^{(3)}(\omega, t) \approx \frac{e^{i\omega t}\, i^{\nu_- - \nu_+}\, (\omega_f t)^{-\xi}}{\nu_f [\Gamma(1 + T\omega_i^{-1})]^{2\xi} (2\pi)^2}
$$
$$
\times \begin{cases} 2^{-2\gamma}\, \Gamma(1 + T\omega_i^{-1})^{2\xi} & t \ll (\epsilon T)^{-1} \\ (T\epsilon t)^{(1 + 2T\omega_i^{-1})\xi}\, (2\pi)^{\xi - 2\gamma} 2^{2\gamma T\omega_i^{-1}} e^{-\pi T\epsilon t\xi} & t \gg (\epsilon T)^{-1} \end{cases}, \quad (45)
$$

while the one for $\tau \to t(\nu_i + \nu_f)\nu_i^{-1}$ is

$$
\mathscr{A}_r^{(4)}(\omega, t) \approx \frac{e^{i\omega t\left(1 + \frac{\nu_f}{\nu_i}\right)}(-i)^{\nu_- - 2\gamma}}{(2\pi)^2 [\Gamma(1 + T\omega_i^{-1})]^{2\xi}} \frac{j(\omega) a^{-1}}{(2\omega_f t)^{\nu_-}}
$$
$$
\times \begin{cases} \Gamma(1 + T\omega_i^{-1})^{2\nu_- - 4\gamma} & t \ll (\epsilon T)^{-1} \\ [2\pi(T\epsilon t)^{(1 + T\omega_i^{-1})}]^{(\nu_- - 2\gamma)} e^{-\pi T\epsilon t(\nu_- - 2\gamma)} & t \gg (\epsilon T)^{-1} \end{cases}. \quad (46)
$$

In the latter equation we have introduced the function

$$
j(\omega) = \int_{-\infty}^{\infty} dz\, e^{i\omega z} \left( \frac{1}{1 + i\omega_f z} \right)^{\nu_+ - 2\gamma} \left| \Gamma(1 + T\omega_i^{-1} + iT\epsilon z) \right|^{\nu_+ - 2\gamma}. \quad (47)
$$

As one can see, $\mathscr{A}_r^{pn}(\omega, t)$ qualitatively behaves like $\mathscr{A}_r^{(2)}(\omega, t)$: at long times $t \gg (\epsilon T)^{-1}$ it becomes negligibly small due to the presence of a fast non-universal exponential decay induced by the finite temperature.

In the end, the transient dynamics of the non-equilibrium spectral function is dominated by

$$
\delta A_r(\omega, t) \equiv A_r^<(\omega, t) - \bar{A}_r(\omega) \approx \begin{cases} (\omega_f t)^{-\xi} & t \ll (\epsilon T)^{-1} \\ -t^{-2}\, \gamma T\omega_i^{-1} \partial_\omega^2 \bar{A}_r(\omega) & t \gg (\epsilon T)^{-1} \end{cases}, \quad (48)
$$

where $\xi$, given in Eq. (43), attains values $1 < \xi \le 2$ for reasonable quenches.

At short times ($t \ll (\epsilon T)^{-1}$) a non-universal power law behaviour with exponent $\xi$ is present and could dominate the relaxation dynamics, masking the universal features. For sufficiently long times $t \gg (\epsilon T)^{-1}$ the situation changes and the non-equilibrium spectral function presents a very clear $t^{-2}$ decay, directly linked to the interaction quench. This is an interesting feature, since in a standard equilibrium case (non-quenched system) one would expect a exponential behaviour for fermionic correlation functions at finite temperature. It is also in contrast with the zero temperature case discussed in Ref. [45] and shown with green curves in Fig. 2 for reference. At $T = 0$, indeed, the non-equilibrium spectral function always shows non-universal power laws $\propto t^{-\xi}$, masking the $\propto t^{-2}$ behaviour. On the contrary, the initial preparation in a thermal state allows to identify of $\propto t^{-2}$ features in the relaxation dynamics: provided that $t \gg (\epsilon T)^{-1}$ this universal power-law decay, a fingerprint of the quench-induced entanglement, remarkably emerges in the non-equilibrium spectral function at long enough times.

All features described so far can be seen in Fig. 2, where the real and the imaginary part of the transient spectral function $\delta A_r(\omega, t)$ are evaluated numerically as a function of $t$ for different temperatures. Notice that we have also inserted the zero temperature case (green line) to better clarify the differences at finite temperature. The universal decay $\propto t^{-2}$ clearly emerges in the real part once the exponential decay of the non-universal contributions sets in, around $t \sim 5(\epsilon T)^{-1}$. A similar behaviour is observed looking at the imaginary part of the non-equilibrium spectral function even if, in this case, the power-law decay is faster and $\propto t^{-3}$.

## 5 Transport properties

In this section we focus on a possible setup to observe the intrinsic features present in the fermionic spectral function induced by the quantum quench. To this end we focus on the relaxation dynamics of the average current injected into the channel from a weakly tunnel-coupled 1D non-interacting probe. It is worth to underline that the probe is a tool to inspect the intrinsic properties of the fermionic channel out-of-equilibrium and, thus, it is supposed to be as non-invasive as possible. A perturbative approach in the weak tunnel coupling is therefore fully justified in evaluating the related transport properties [61].[1] Fermions of the probe are described by the Hamiltonian

$$H_p = -iv \int_{-\infty}^{+\infty} dx \, \chi^\dagger(x)\partial_x\chi(x), \tag{49}$$

with $\chi(x)$ the fermionic field. The probe is initially ($t < 0$) prepared in equilibrium at a given temperature $T_p$ (not necessarily equal to $T$) with associated density matrix $\rho_{eq}^{(p)} = e^{-H_p/T_p}/Z_p$ with $Z_p = \mathrm{Tr}[e^{-H_p/T_p}]$. It is also subjected to a bias voltage $V$ taken with respect to the Fermi level of the channel.

A localised injection at $x = x_0$ is switched on at $t = 0^+$ – immediately after the quench – and is described by the tunneling Hamiltonian [44, 45, 63–65]

$$H_t(t) = \vartheta(t)\lambda \sum_{r=L,R} \psi_r^\dagger(x_0)\chi(x_0) + \mathrm{H.c.}, \tag{50}$$

with $\lambda$ the tunneling amplitude and $\vartheta(t)$ the step function. We note that after the tunneling process both the channel and the probe will evolve dynamically and can be considered as a closed system. We are interested into the total injected current $I(V,t) = I_+(V,t) + I_-(V,t)$, where

$$I_\eta(V,t) = q \, \partial_t \int_{-\infty}^{+\infty} dx \, \langle \delta n_\eta(x,t) \rangle \tag{51}$$

is the chiral current associated to the $\eta$ channel, with $q$ the fermion charge. Here,

$$\langle \delta n_\eta(x,t) \rangle = \mathrm{Tr}\{n_\eta(x,t)[\rho_{tot}(t) - \rho_{tot}(0)]\}, \tag{52}$$

where $\rho_{tot}(0) = \rho_{eq} \otimes \rho_{eq}^{(p)}$ and its time evolution is computed in the interaction picture with respect to $H_t$. The chiral particle density reads

$$n_\eta(x,t) = -\eta\sqrt{\frac{K_f}{2\pi}}\partial_x\phi_{f,\eta}(x - \eta v_f t). \tag{53}$$

At the lowest order in the tunneling amplitude, the total current can be written as [45]

$$I(V,t) = 2q|\lambda|^2 \mathrm{Re}\left\{ i \int_0^t d\tau \left[ \sum_{r=L,R} G_r^<(t, t-\tau) \right] G_p^>(\tau) \sin(qV\tau) \right\}, \tag{54}$$

where $G_r^<(t, t-\tau)$ is the lesser Green function of the fermionic channel defined in Eq. (13) and

$$G_p^>(\tau) = -\frac{i}{2\pi a} \frac{\left| \Gamma(1 + T_p\omega_p^{-1} + iT_p\tau) \right|^2}{\Gamma^2(1 + T_p\omega_p^{-1})} \frac{1}{1 - i\omega_p\tau} \tag{55}$$

---

[1]Indeed, a weak and point-like tunneling between the probe and the 1D channel does not affect the latter [61]. Note that a perturbative approach would be justified also for more generic tunnel couplings as long as they are weak enough and there is a finite energy scale set by finite (effective) temperature and bias [49, 61, 62].

is the greater Green function of the probe with $\omega_p = v/a$ the associated energy cut-off. Recalling that the lesser Green function can be expressed as $G_r^<(t, t-\tau) = \int_{-\infty}^{+\infty} e^{-i\omega\tau} A_r^<(\omega, t) d\omega$ by Fourier transform, one immediately recognise the direct link with the non-equilibrium spectral function of the channel discussed in the previous Section.

Plugging Eq. (21), with $D_{\alpha,\beta}(t,\tau)$ given in Eqs. (24-27), and Eq. (55) into Eq. (54) one can eventually write the total current as

$$
\begin{aligned}
I(V, t) = \frac{4q|\lambda|^2}{(2\pi a)^2} \, \mathrm{Re} \Bigg\{ &\int_0^t d\tau \; \frac{\left|\Gamma(1 + T_p\omega_p^{-1} + iT_p\tau)\right|^2}{\Gamma^2(1 + T_p\omega_p^{-1})} \frac{1}{1 - i\omega_p\tau} \\
&\times \left[ \frac{\left|\Gamma(1 + T\omega_i^{-1} + iT\epsilon\tau)\right|^2}{\Gamma^2(1 + T\omega_i^{-1})} \right]^{\nu_+ + \nu_-} \left[ \frac{1}{1 - i\omega_f\tau} \right]^{\nu_+} \left[ \frac{1}{1 + i\omega_f\tau} \right]^{\nu_-} \\
&\times \left[ \frac{(1 + 4\omega_f^2(t-\tau)^2)(1 + 4\omega_f^2 t^2)}{(1 + \omega_f^2(2t-\tau)^2)^2} \right]^{-\gamma} \\
&\times \left[ \frac{\left|\Gamma(1 + T\omega_i^{-1} - iT\epsilon(2t-\tau))\right|^2}{\left|\Gamma(1 + T\omega_i^{-1} + 2iT\epsilon(t-\tau))\right| \left|\Gamma(1 + T\omega_i^{-1} + 2iT\epsilon t)\right|} \right]^{-4\gamma} i\sin(qV\tau) \Bigg\},
\end{aligned}
\tag{56}
$$

where $\nu_\pm$ and $\gamma$ are defined in Eqs. (36) and (39), respectively. Equation (56) contains both the initial temperatures of the probe $T_p$ and of the channel $T$.

Useful analytical expansions in the long-time limit can be obtained following the same procedure described in the previous Section. In particular, in the long-time limit the current decomposes into two terms

$$
I(V, t) \approx I^{(1)}(V, t) + I^{(2)}(V, t),
\tag{57}
$$

the former stemming from an expansion of the integrand for $\tau \to 0$ and the latter from an expansion for $\tau \to t$ [45]. $I^{(1)}(V, t)$ is responsible for the universal features and reads

$$
I^{(1)}(V, t) = \bar{I}(V) - \frac{1}{t^2} \mathscr{F}(V, t),
\tag{58}
$$

where

$$
\mathscr{F}(V, t) = \kappa(t) \frac{\gamma}{q^2} \partial_V^2 \bar{I}(V),
\tag{59}
$$

with $\kappa(t)$ given in Eq. (39). The steady-state current is

$$
\begin{aligned}
\bar{I}(V) = \frac{4q|\lambda|^2}{(2\pi a)^2} \mathrm{Re} \Bigg\{ &\int_0^t d\tau \; \frac{1}{1 - i\omega_p\tau} \left( \frac{1}{1 - i\omega_f\tau} \right)^{\nu_-} \left( \frac{1}{1 + i\omega_f\tau} \right)^{\nu_+} \\
&\times \frac{|\Gamma(1 + T_p\omega_p^{-1} + iT_p\tau)|^2}{\Gamma^2(1 + T_p\omega_p^{-1})} \left[ \frac{|\Gamma(1 + T\omega_i^{-1} + iT\epsilon\tau)|^2}{\Gamma^2(1 + T\omega_i^{-1})} \right]^{\nu_+ + \nu_-} i\sin(qV\tau) \Bigg\}.
\end{aligned}
$$

Looking at the term $I^{(2)}(V, t)$, it is useful to introduce $z = t - \tau$ and expand the integrand of

Eq. (56), retaining only the leading term in $z/t$. One thus obtains

$$I^{(2)}(V,t) = \frac{\Phi(V,t)\, 2^{-2\gamma}}{(\omega_f t)^{1+\nu_+ + \nu_- - 2\gamma}} \left[ \frac{\left|\Gamma(1 + T\omega_i^{-1} - iT\epsilon t)\right|^4}{\left|\Gamma(1 + T\omega_i^{-1} + 2iT\epsilon t)\right|^2} \right]^{-2\gamma} \left[ \frac{\left|\Gamma(1 + T\omega_i^{-1} + iT\epsilon t)\right|^2}{\Gamma^2(1 + T\omega_i^{-1})} \right]^{\nu_+ + \nu_-}$$

$$\times \frac{\left|\Gamma(1 + T_p\omega_p^{-1} + iT_p t)\right|^2}{\Gamma^2(1 + T_p\omega_p^{-1})}, \quad (60)$$

where

$$\Phi(V,t) = \frac{-4q|\lambda|^2}{(2\pi a)^2 K_f} \cos\left[\frac{\pi}{2}(\nu_+ - \nu_-)\right] \int_0^{+\infty} dz\, \frac{|\Gamma(1 + T\omega_i^{-1} + 2iT\epsilon z)|^{4\gamma}}{[1 + (2\omega_f^2 z^2)]^\gamma} \sin[qV(t-z)] \quad (61)$$

is an oscillating function of time with period $\propto (qV)^{-1}$. Exploiting the asymptotic expression for the gamma functions in Eq. (30), one finds

$$I^{(2)}(V,t) \approx \Phi(V,t) \begin{cases} \left[2\Gamma(1 + T\omega_i^{-1})\right]^{-2\gamma} (\omega_f t)^{-\xi-1} & t \ll (\epsilon T)^{-1}, T_p^{-1} \quad (62a) \\[2mm] \left[2\Gamma(1 + T\omega_i^{-1})\right]^{-2\gamma} \varphi_p(T_p,t)\,(\omega_f t)^{-\xi-1}\, e^{-\pi T_p t} & T_p^{-1} \ll t \ll (\epsilon T)^{-1} \quad (62b) \\[2mm] \varphi(T,t)\,(\omega_f t)^{-\xi-1}\, e^{-\pi T\epsilon t(\nu_+ + \nu_-)} & (\epsilon T)^{-1} \ll t \ll T_p^{-1} \quad (62c) \\[2mm] \varphi(T,t)\,\varphi_p(T_p,t)\,(\omega_f t)^{-\xi-1}\, e^{-\pi T\epsilon t(\nu_+ + \nu_-) - \pi T_p t} & (\epsilon T)^{-1}, T_p^{-1} \ll t. \quad (62d) \end{cases}$$

Here,

$$\varphi_p(T_p,t) = 2\pi \left[\Gamma(1 + T_p\omega_p^{-1})\right]^{-2} (T_p t)^{1 + 2T_p\omega_p^{-1}}, \quad (63)$$

$$\varphi(T,t) = (2\pi)^\xi\, 2^{4\gamma T\omega_i^{-1}} \left[\Gamma^2(1 + T\omega_i^{-1})\right]^{-\nu_+ - \nu_-} (T\epsilon t)^{\xi(1 + 2T\omega_i^{-1})} \quad (64)$$

are power-law corrections to the exponential decays in time. As one can see $I^{(2)}(V,t)$ contains all the non-universal contributions.

To discuss the analytical results obtained so far, we begin by recalling the $T = T_p \to 0$ limit. In this case Eq. (58) with $\kappa(t) = -1/2$ and Eq. (62a) contribute to $I(V,t)$: the current decay shows then a competition between the $t^{-2}$ power law and a non-universal power law $\propto t^{-\xi-1}$. Since for reasonable quenches $\xi + 1 \gtrsim 2$, the two power laws are similar: although the universal decay is leading, distinguishing its contribution in $I(V,t)$ can be cumbersome. Let us now consider the case of a thermal preparation for the channel and the probe starting with the case $T = T_p > 0$, considering quenches which increase the interaction strength, i.e. $K_f < K_i$ (implying $\epsilon > 1$). Here, two different time regimes must be distinguished:

- For $t \ll (\epsilon T)^{-1}$, the same competition between the universal $\propto t^{-2}$ and the non-universal $\propto t^{-\xi-1}$ power laws as in the $T = 0$ case is present;

- For $t \gg (\epsilon T)^{-1}$, Eq. (58) with $\kappa(t) = T\omega_i^{-1}$ and Eq. (62c) (for $t \ll T_p^{-1}$) or Eq. (62d) (for $t \gg T_p^{-1}$) contribute to the current. In both cases, the non-universal power law is now exponentially suppressed and the decay of $I(V,t)$ is *entirely* dominated by a universal power law $t^{-2}$ with a prefactor $\propto T$.

Thus we can conclude that a thermal preparation with a common temperature $T$ for the system strongly enhances the visibility of the universal decay, setting a time scale $(\epsilon T)^{-1}$ beyond which the latter becomes the only relevant contribution to $I(V,t)$.

The situation is even better if the case $T_p > \epsilon T > 0$ is considered. Now, three regimes arise

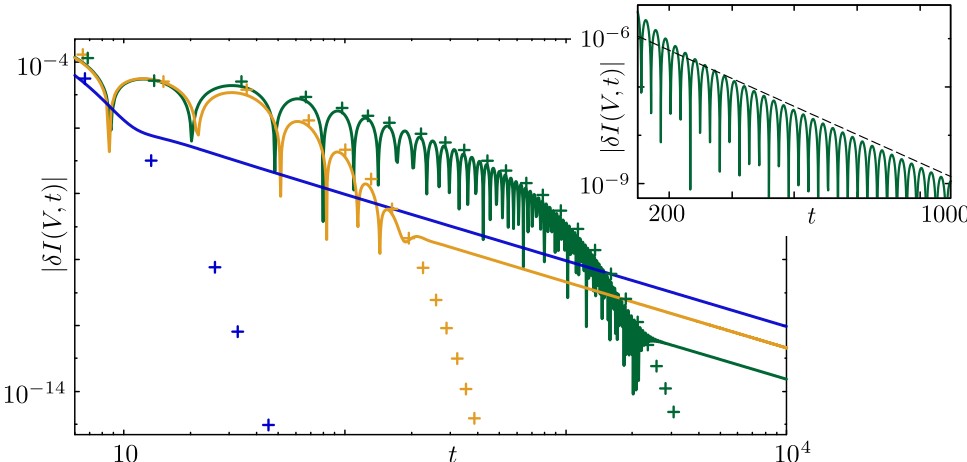

Figure 3: Solid lines show $|\delta I(V,t)|$ (units $4q|\lambda|^2\,\omega_f^{-1}a^{-2}$) as a function of time in a quenched channel with $K_i = 0.9$ and $K_f = 0.6$. Crosses show $|\delta I(V,t)|$ in a non-quenched channel with $K_i = K_f = 0.6$. Different colors refer to different temperatures $T = T_p$: $T = 10^{-1}$ (blue), $T = 10^{-2}$ (orange), $T = 10^{-3}$ (green). The inset displays $|\delta I(V,t)|$ in the absence of quench with $K_i = K_f = 0.6$ and $T = 10^{-3}$ on a logarithmic plot. The black dashed line shows an exponential decay $\propto \exp[-\pi T\epsilon t(1 + \nu_+ + \nu_-)]$. Here $qV = 0.1\omega_f$, time is in units $\omega_f^{-1}$, and temperature in units $\omega_i$.

- For $t \ll T_p^{-1}$, both the universal $\propto t^{-2}$ and the non-universal $\propto t^{-\xi-1}$ power laws compete;

- For $T_p^{-1} \ll t \ll (\epsilon T)^{-1}$, Eq. (58) with $\kappa(t) = -1/2$ and Eq. (62b) characterise $I(V,t)$: the non-universal power law is exponentially suppressed and $I(V,t)$ decays universally $\propto t^{-2}$;

- For $t \gg (\epsilon T)^{-1}$, Eq. (58) with $\kappa(t) = T\omega_i^{-1}$ and Eq. (62d) must be considered for $I(V,t)$: the non-universal power law is still exponentially suppressed and the decay of $I(V,t)$ is still led by the universal decay with a prefactor $\propto T$.

Thus, by suitably choosing a larger temperature $T_p$ for the initial preparation of the probe with respect to the initial temperature $T$ of the channel, the universal power law is the only leading term beyond the time scale $T_p^{-1}$ and two regimes with universal decay but different prefactors are now visible (see the last two points above).

We close this discussion by commenting the non quenched case ($\gamma = 0$, $\epsilon = 1$) with $T_p > \epsilon T > 0$: without quench the universal power is not present - see Eq. (58). For $t \ll T_p^{-1}$ the non-universal power law $\propto t^{-\xi-1}$ is found, while for $t \gg T_p^{-1}$ the current decays exponentially as normally expected. Thus, the presence of a universal decay is a unique fingerprint of the quantum quench, stable even in the case of a preparation at a finite temperature.

All these features are confirmed by the numerical evaluation of Eq. (56). In Fig. 3 (main panel) the quantity $|\delta I(V,t)| = |I(V,t) - \bar{I}(V)|$ is plotted as a function of $t$, on a bi-logarithmic scale, for different initial temperatures of the whole setup, with $T = T_p$. The two regimes $t \ll (\epsilon T)^{-1}$ and $t \gg (\epsilon T)^{-1}$ are clearly visible: the former features oscillations as a result of the strong competition between the universal power-law decay and the non-universal one which, for the quench considered here, has a non-universal exponent $\simeq -2.3$. By contrast, the latter regime shows a very clean decay $\propto t^{-2}$, resulting from the exponential suppression of the non-universal oscillating terms. The crosses in Fig. 3 (main panel) show $|\delta I(V,t)|$ in

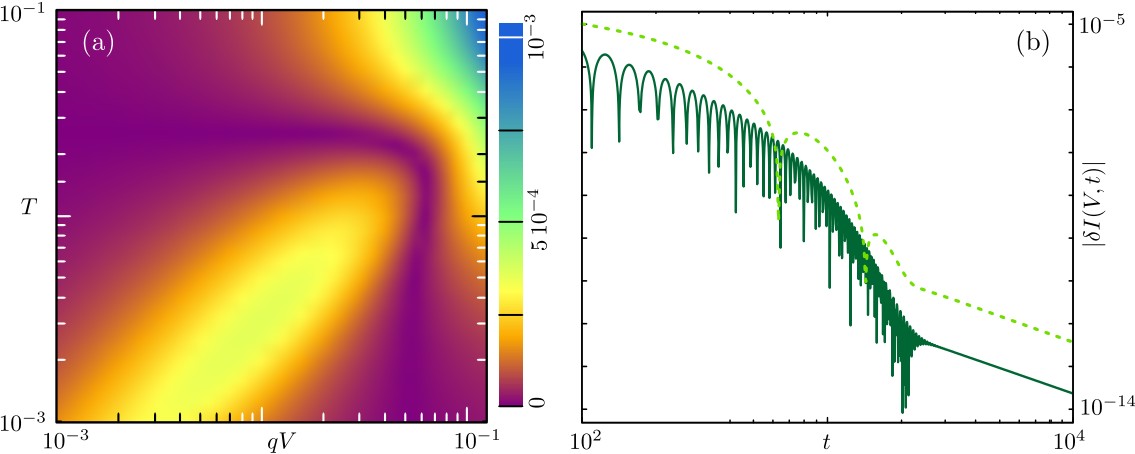

Figure 4: (a) Density plot of $|\mathcal{F}(V,T)|$ (units $4q\gamma|\lambda|^2\,\omega_f^{-3}a^{-2}$) as a function of the bias voltage $qV$ (units $\omega_f$) and the temperature $T$ (units $\omega_f$). (b) Plot of the transient current $|\delta I(V,t)|$ (units $4q|\lambda|^2\omega_f^{-1}a^{-2}$) as a function of time, for temperature $T = T_p = 10^{-3}\omega_i$ and for two different bias voltage: $qV = 0.1\omega_f$ (solid line) and $qV = 0.004\omega_f$ (dotted line). In both Panels, the quench is from $K_i = 0.9$ to $K_f = 0.6$.

the non-quenched case with $K_i = K_f$: the fast exponential decay for $t \gg (\epsilon T)^{-1}$ is evident (note that, to improve the readability of the Figure, the data sampling rate is kept very low). A more detailed plot of the non-quenched current is shown with a logarithmic scale in the Inset, where the oscillations of the non-universal decay can be clearly seen enveloped by an exponential decay, which is in good agreement with Eqs. (62c,62d) - see the dashed black line.

The $t \gg (\epsilon T)^{-1}$ regime is the most interesting one in order to assess the different time evolution of quenched and non-quenched channels at finite temperature with $T = T_p$. It is therefore worth to discuss how the prefactor $\mathcal{F}(V,T)$ of the universal power law, introduced in Eqs. (58) and (60), depends on the temperature $T$ and on the bias $V$ in this regime. In Fig. 4(a), we plot $|\mathcal{F}(V,T)|$ focusing on a reasonable range of the parameters. Interestingly, this function turns out to be non-monotonic with respect to both $T$ and $V$. This fact leads to a non-trivial feature: considering temperatures lower than $10^{-2}\omega_i$, a decrease of the voltage bias can actually determine an increase by orders of magnitude of the prefactor $|\mathcal{F}(V,T)|$. As an example of this behaviour, in Fig. 4(b) we plotted the current $|\delta I(V,t)|$ as a function of time for two different voltage biases but with the same temperature $T = 10^{-3}\omega_i$. In the $t \gg (\epsilon T)^{-1}$ regime, the dotted line ($qV = 0.004\omega_f$) lies more than one order of magnitude above the solid one ($qV = 0.1\omega_f$). Therefore, a proper tuning of the voltage bias can be useful to magnify and to detect the peculiar features induced by the presence of the quantum quench.

In Fig. (5), we show $|\delta I(V,t)|$ for the case $T_p \gg \epsilon T > 0$. The two regimes $T_p^{-1} \ll t \ll (\epsilon T)^{-1}$ with $|\delta I(V,t)| \propto 1/(2t^2)$ and $t \gg (\epsilon T)^{-1}$ with $|\delta I(V,t)| \propto T/(\omega_i t^2)$ are clearly evident, as well as the different prefactors that characterise them. This confirms that preparing the probe in a state with a higher temperature strongly enhances the visibility of the universal decay of the current.

## 6    Conclusions

We have investigated the relaxation dynamics of a 1D channel of spinless interacting fermions, initially prepared in a thermal state at $T > 0$, subject to a sudden quench of the interaction strength. The channel is modeled as a Luttinger liquid. The out-of-equilibrium fermionic spec-

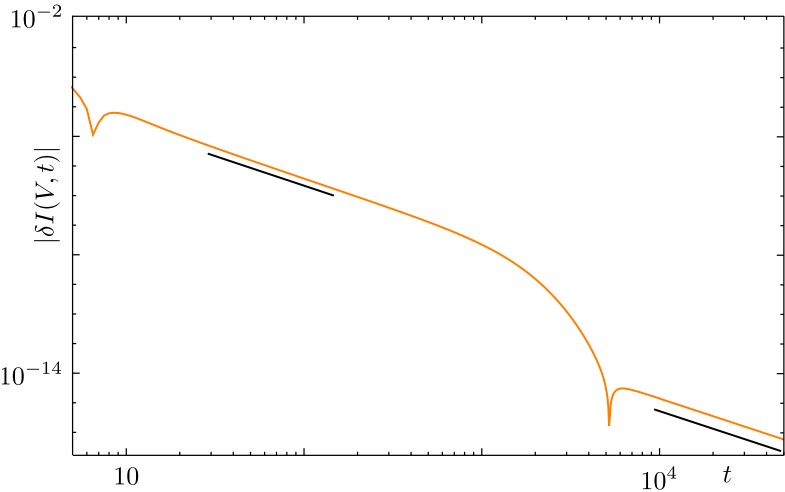

Figure 5: Plot of $|\delta I(V,t)|$ (units $4q|\lambda|^2 \omega_f^{-1} a^{-2}$) as a function of time with $K_i = 0.9$ and $K_f = 0.6$. The temperatures of the channel and of the probe are $T = 2 \cdot 10^{-4} \omega_i$ and $T_p = 2 \cdot 10^{-1} \omega_i$, respectively. The black lines shows decays $\propto t^{-2}$. Here $qV = 0.1\omega_f$, time is in units $\omega_f^{-1}$.

tral function is analytically evaluated and studied in details. One of the main results is that it shows a universal $t^{-2}$ decay towards the steady state in presence of finite temperature, showing the peculiar role associated to the quantum quench and its link with universal power-law decay. Indeed, at finite temperature, non-universal factors acquire an exponential decay, greatly enhancing the visibility of the decay $t^{-2}$ in the long time limit. This is in sharp contrast with the expectation in absence of an interaction quench, where fermionic correlation functions would acquire an exponential behaviour due to the presence of a finite temperature. These features are also reflected in observable quantities such as charge current. In particular, we have analysed the charge current flowing to the channel from a locally tunnel-coupled biased probe, prepared before the quench in a thermal state with a temperature $T_p$. A weak tunneling is switched on immediately after an interaction quench, and the evolution of the closed system composed by channel and probe is analysed to lowest order in the tunneling amplitude. The current decays towards its steady value with the same universal power law $\propto t^{-2}$ which characterises the universal relaxation of the non-equilibrium spectral function. Not only the persistence of this universal decay confirms the robustness of the results previously obtained in the $T \to 0$ limit [45], but finite temperature can be a tool to enhance the visibility of this universal scaling induced by the quench. Finally we have shown that, by acting on the initial temperature of a tunnel-coupled probe, the visibility of the $\propto t^{-2}$ decay can be enhanced even further.

For a typical channel of cold atoms [4], one can estimate temperatures of about $T \sim 100$ nK, for which the thermal time-scale is about $T^{-1} \sim 500$ $\mu s$. Typical experiments in such a system can explore time domains well in excess of the hundred of ms, so it seems feasible, at least in principle, to observe the striking differences between quenched and non-quenched systems in the presence of thermal effects.

## Acknowledgements

A. C. and T. L. S. acknowledge support from the National Research Fund Luxembourg (AT-TRACT 7556175). M. C. acknowledges support from the CNR-CONICET cooperation pro-

gramme "Energy conversion in quantum, nanoscale, hybrid devices". A. C., F.M. G., F. C. and M. S. gratefully acknowledge support from UNIGE - Genova University.

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
