# Peer review of "Universal scaling of quench-induced correlations in a one-dimensional channel at finite temperature"

_SciPost Physics, doi:SciPost Phys. 4, 023 (2018)_

## Round 1 · Referee Report · Anonymous · 2017-12-5

Strengths

see report

Weaknesses

see report

Report

The questions of if and how observables and correlation functions
of closed one-dimensional quantum many-body systems relax towards
a steady state value after an abrupt change of the amplitude of the
two-particle interaction are currently heavily addressed in model
studies. It is furthermore of interest to clarify if the long-time
asymptotic expectation values, if reached, can be computed using
a properly chose ensemble.

The authors of the present manuscript consider the Tomonaga-Luttinger
(TL) model. The field-theoretical variant is studied, which means that
the Hamiltonian Eqs. (1) and (2) is ultraviolet divergent as written.
To obtain finite results for correlation functions an ad hoc
regularization of integrals must be introduced "by hand". This
procedure should be considered as part of the model, is, however,
left implicit. This is often done when using phenomenological
bosonization (as opposed to constructive bosonization).

The authors first compute the time evolution of the single-particle
lesser Green function with respect to the TL Hamiltonian with a
final interaction strength. The starting state is the thermal
equilibrium one (initial temperature T>0) with respect to an
initial interaction. Only the case with equal positions of the two
fermionic fields entering the Green function is considered. As
their main result the authors find that the correlation between
the left and right moving chiral bosonic fields decays as t^{-2}
with the absolute time t after the interaction quench. This cross
correlator enters the expression for the Green function which
thus features a term showing the same type of long-time decay.
Similar calculations were performed in many earlier publications,
e.g. in Ref. [41] by (a subclass of) the present authors themselves.
It is e.g. well established that in the long-time limit the
expectation values of local observables and correlation functions
of the TL model can be understood in terms of a generalized
Gibbs ensemble (GGE). I am in this respect puzzled by the authors
statement that the GGE of the TL model is "...characterized by an
infinite number of local conserved quantities..." (see the second
sentence of the third paragraph on page 2). The conserved mode
occupancies naturally appearing in the GGE are spatially non-local.
Although the GGE description might not be unique I am not aware that
for the TL model a GGE build out of spatially local conserved
quantities was constructed.

In Ref. [41] the same model and quench protocol was discussed but
for initial temperature T=0 leading to the same result for the cross
correlator decay and thus the Green function as found in the present
manuscript for T>0. As discussed in Ref. [41] in the non-equilibrium
spectral function, a reasonable measurable quantity derived from the
Green function by Fourier transform with respect to the relative
time of the two fields, this "universal" decay is masked by other
terms. These show typical Luttinger liquid power-law decay in t
with interaction dependent exponents which turn out to be generically
larger than -2. Surprisingly, this problem is not even mentioned in
the present manuscript. However, up to this point, that is Section 3,
the setup is transparent, the calculations are simple and
conceptually straightforward.

To construct an observable which reveals the above "universal" t^{-2}
decay the authors suggest a modified setup. In this the one-dimensional
(1D) interacting system is at the quench time t=0 locally tunnel coupled
to a (chiral 1D) probe (reservoir) system. The total current injected
into the system is computed using lowest-order perturbation theory in
the tunnel coupling; the expression Eq. (30) for the current then
involves the lesser Green function of the isolated 1D system. Besides
this Eq. (30) contains the greater Green function of the isolated probe.
In Eq. (31) the authors give an analytic expression for this. I am puzzled
that via \omega_f and K_f this (non-interacting) Green function contains
information about the interaction strength in the 1D system? In fact,
in Eq. (A5) of Ref. [41], a publication which already contains the idea
of the modified setup, the authors present an expression for the greater
Green function of the isolated probe (in this case for T=0) which is
independent of the interaction in the system. To me this appears to be
more reasonable. Can the authors comment on this?

After modifying the setup by including the probe a conceptual
difficulty arises. With the coupling of the 1D interacting system
to the infinite probe reservoir held in thermal equilibrium the
authors no longer consider an isolated quantum system but rather
an open one. One would thus expect that the system asymptotically
reaches the equilibrium thermal state imprinted by the reservoir
and not the "closed system" state for which local observables can be
computed employing the non-equilibrium GGE. One would furthermore
expect that the approach towards this equilibrium state is dominated
by an exponential time dependence with rates set by the reservoir-system
coupling (and modified by temperature). In that sense the system
dynamics is heavily affected by the reservoir. However, this change of
the entire dynamics is neither mentioned in the present manuscript
(or Ref. [41] for that matter) nor does it seem to be producable by
the authors perturbative approach to the tunnel coupling. I thus
suspect that what the authors hope for is the following: The time
scales are sufficiently separated such that first the GGE "closed system"
state develops (with the t^{-2} behavior of the cross correlator as
computed for the closed system) while the coupling to the reservoir
affects the dynamics only at a well separated later stage. If my
suspicion is correct the authors must provide arguments that hoping
for this type of time scale separation is reasonable in realistic
setups. On what time scales can one expect to detect the behavior
discussed in Ref. [41] and the present manuscript? Is this a reasonable
scale in the light of cold Fermi gas experiments? Can one perform
improved calculations which show that such simple estimates are
reasonable? If I am mistaken the authors must provide an alternative
way how to circumvent this conceptual problem of using closed system
results in an open system setup. In any case I am very much surprised
that the authors do not explicitly mention this type of conceptual
difficulty.

The coupling to the probe in addition induces a local inhomogeneity
to the Luttinger liquid which might affect the dynamics. It is well
established that local inhomogeneities strongly change the equilibrium
low-energy physics of Luttinger liquids. Can the authors exclude that
this is an issue in the non-equilibrium dynamics of the suggested
setup as well? Again the computational tool, namely perturbation theory
in the system-reservoir coupling might be insufficient to capture
and/or detect the proper impurity physics. The quench studied is not
only one of the global interaction but at the same time a local
single-particle parameter is changed (tunneling). Quenches of local
parameters in the TL model (and related lattice models) were studied
earlier. These studies might provide guidance for what to expect in
the present case.

Even if one ignores the above issues for the moment one might be
tempted to conclude that the progress presented in the present
manuscript (t^{-2} decay of the cross correlator for T>0) is rather
small as compared to what (a subclass of) the authors already
reported on in Ref. [41] (t^{-2} decay of the cross correlator
for T=0). Can the authors make a stronger point why the extension
of the T=0 result justifies another publication?

After the authors have properly responded to all the issues raised
above I am very much willing to reconsider my current decision to
not recommend publication.

Requested changes

see report

---

## Round 1 · Referee Report · Anonymous · 2017-12-10

Strengths

Timely subject.

Weaknesses

Just a small progress with respect to what have been already presented in Ref.[41] by the authors.

Report

In this paper the authors analyse the relaxation dynamics of a Tomonaga-Luttinger model
after preparing the system in a thermal initial state and suddenly quenching the interaction strength.
In the filed theory, after bosonization, and provided a careful regularisation of the ultraviolet divergencies
(which is encoded into the short-length cutoff “a”), all the calculations can be easily push forward
thanks to the quadratic nature of the bosonic Hamiltonians (cf. eq. (6) and (7)). Indeed, the quench
simply reduces to a sudden change of the Luttinger Liquid parameter “K”.

The authors focus on the diagonal part of the dynamical green function
(the two-point Fermionic correlation function at different times and equal positions).

The main result in this regard is the large-time asymptotic decay (t^-2) which confirms
what has been already found in Ref. [41] for a zero temperature initial state, providing evidence that
temperature does not modify the leading time-dependent contribution into the approaching the stationary state.
The authors stress the fact that this contribution arises from the bosonic cross correlation term which
indeed is related to the quench protocol. Honestly, I’m not much surprised by the fact that the
cross correlation survives even for a thermal initial state, since it is merely a consequence of the
quench protocol.

Regarding this section (namely Sec. 3), I think it’s clearly written, nevertheless,
I was struggling by figuring out how the (\tau/2t)^2 behaviour in Eq.(24)
for \tau << t << T^-1 comes from Eq. (22). Moreover, I would like to draw the attention of
the authors to a strongly related result about the low-energy description of an interaction quench
in the XXZ spin chain (Phys. Rev. B 92, 125131 (2015)).

Let me now comment about Sec. 4. Here the authors propose to investigate the
relaxation dynamics by looking at the transport properties arising by injecting fermions into the system.
They thus engineer a noninteracting probe locally coupled (for t>0 at x=x0) to the original Hamiltonian.
They therefore analyse the total particle current. This calculation
having been done at the first order in perturbation theory with respect to
the probe-system coupling strength. The authors state that the
fermionic field of the probe “chi(x)” is kept at fixed temperature “T”. Now I’m a bit confused:
(1) is T the same temperature at which the original system has been prepared?
What about different temperatures?
(2) When the authors claim that the probe is at thermal equilibrium, what do they exactly mean?
In other words, I suppose the probe field is a new dynamical variable of the new setup,
which evolves according to the new post-quench Hamiltonian. Is this the case?
Otherwise, if the probe is really kept at fixed temperature, then in the new setup,
the system is no longer a closed system. Therefore, although there could be an intermediate regime
for which the system relaxes toward a generalised thermal ensemble, at very large time, due to
the influence of the external bath, I expect the system thermalising. Maybe thermalisation occurring
starting from x_0, with a sort of light-cone effect. Can the authors be more clear about this.

In particular, I’m really curious about the effect of the new setup regarding the “local” quench with
the probe. Indeed, as far as the global quench is joined with a local quench, I expect that, on top of the
homogeneous dynamics induced by the global quench, there should be a sort of spreading of particles density injected in x_0.
This leading to two different stationary descriptions, inside and outside the light cone.

To conclude, even though I appreciated this work, I can support publication only after
Sec. 4 has been largely rewritten in order to address all the points I mentioned before
so as to clarify the setup.

Requested changes

see report.

---

## Round 2 · Referee Report · Anonymous (Referee 2) · 2018-3-12

Report

After the author's revision, I think the manuscript has been sufficiently improved.
I particularly appreciated the fact that the authors clarified the protocol involving the probe.
Even though I'm still not fully convinced about the novelty of the results
(especially with respect to what has been already presented by the authors in a previous publication),
they nevertheless presented a generalisation to thermal initial states which may deserve to be published.

---

## Round 2 · Referee Report · Anonymous (Referee 1) · 2018-3-28

Report

I first would like to apologize that it took me unexpectedly long to study the revised version and prepare the present report.

In response to the first round of referee reports the authors made two major changes.

  1. To enhance the relevance of the work which was questioned by both referees they added a chapter on the absolute time-dependence of the local "spectral function" (Fourier transform of the local Green function with respect to the relative time). This indeed provides further insights. However, one might argue that the chapter on the transport setup (see 2.) is now superfluous. With respect to this new chapter I am puzzled by the first line of Eq. (35). While on the left hand side the limit t -> oo was taken, t appears explicitely on the right hand side!?

  2. The authors made an effort to now explain more clearly the transport setup. Both referees were confused by the original description. However, I am still not fully convinced that the field theory the authors study can be realized in any microscopic model (not to speak of experiments). In PRL 105, 266404 mentioned in the authors reply it is emphasized that "...we have shown that the picture of tunneling into a LL is qualitatively modified when the tunneling amplitude is not treated as infinitesimally small. The conventional FP [fixed point] has a finite basin of attraction only in the model of the point tunnel contact, but taking a finite size of the contact (or any perturbation induced by the contact in the wire) into account makes it unstable." If the "conventional FP" is generically unstable (as stated in this paper) the probe will generically be invasive and the physics on asymptotic time scales will not be the one of the unperturbed system (the probe does not simply probe the system). In any case the authors revisions are in this respect inappropriate:

"It is worth to underline that the probe is a tool to inspect the intrinsic properties of the fermionic channel out-of-equilibrium and, thus, it is supposed to be as non-invasive as possible. A perturbative approach in the weak tunnel coupling is therefore fully justified in evaluating the related transport properties."

This paragraph does not contain any arguments that a proper "non-invasive" limit exists at all. The latter is the question of utter relevance (see my quote from the paper brought up by the authors). At the minimum the authors must properly revise the wording and backup their view by the proper references.

However, as emphasized above with the new chapter on the spectral function one might argue that the chapter on the transport setup even lost its relevance. The role of the transport setup was to my understanding only to illustrate the mathematical findings in a reasonable observable (current). The spectral function can now take over this role. I suggest that the authors consider to simply remove the transport chapter.

To summarize, I am still not convinced that the manuscript contains highly exciting new results (refer to the earlier paper by (almost) the same authors) and doubt that the (field theoretical) transport considerations will withstand in microscopic models (again not to speak of experiments). However, in my view both holds for many other publications in the field of quantum quenches. I will thus not oppose publication.

---

## Round 2 · Author Response

First of all, we take the opportunity to thank both Referees for their careful reading of our manuscript. Their comments and criticism helped us to improve the quality of our article, clarifying our findings and the novelty of the presented results. The second Referee has an overall positive view of our work, stating "I appreciated this work" although his recommendation is to revise Sec. 4 to clarify some points. Even the first Referee states that he is "very much willing to reconsider" his decision about our paper once we have answered to the issues he raised.

In revising the work, we have taken into account all the questions raised by both Referees. In particular, we have considered -- and addressed -- their common criticism about the description of the coupling with the probe. We now specify with much greater detail that the channel and the probe constitute a closed system, whose global dynamics is considered after an interaction quench. In the revised version, we also underline with more emphasis and clarity our main result, namely the persistence of a universal power law in the decay of fermionic properties of the channel at finite temperature. We believe that the introduction of a new section on the fermionic properties and their relaxation dynamics makes the results more transparent, clarifying some misunderstanding of the previous version.
To do so, following the suggestion of Report 1, we have added an entirely new Section about the behavior of the non-equilibrium spectral function at finite temperature. Moreover, in the Section devoted to the study of the transport properties of the quenched channel, we now provide new results for the more general case when the channel and the probe are initially prepared at different temperatures.

Besides the new material described above, the revision has required to re-write several points of the manuscript including the abstract, parts of the introduction and of the main text. After this revision we are convinced that the results of our work are more clear and the overall quality of the manuscript has improved, and therefore hope for a positive conclusion of the referral process.

In the following, we reply in details to both Referees concerning all the questions/criticisms raised. In the last part of this letter, a list of changes is also provided.
With best regards,

Alessio Calzona on behalf of all the Authors.
* * *
REPORT 1

R1: "The authors of the present manuscript consider the Tomonaga-Luttinger (TL) model. The field-theoretical variant is studied, which means that the Hamiltonian Eqs. (1) and (2) is ultraviolet divergent as written. To obtain finite results for correlation functions an ad hoc regularization of integrals must be introduced "by hand". This procedure should be considered as part of the model, is, however, left implicit. This is often done when using phenomenological bosonization (as opposed to constructive bosonization)."
A1: The Referee is right, we explicitly regularize the field theory by means of a (standard) exponential cutoff and thus do not adopt constructive bosonization. We have added a sentence in the paper specifying this point, also citing now two new references concerning the constructive bosonization.

R2: "I am in this respect puzzled by the authors statement that the GGE of the TL model is "...characterized by an infinite number of local conserved quantities..." (see the second sentence of the third paragraph on page 2). The conserved mode occupancies naturally appearing in the GGE are spatially non-local. Although the GGE description might not be unique I am not aware that for the TL model a GGE build out of spatially local conserved quantities was constructed."
A2: We do agree with the Referee. Indeed, our GGE has been built out of the non-local mode occupation numbers of the post-quench Hamiltonian. In the revised version we have removed the reference to "local conserved quantities" in order to avoid confusion. However, we wish to point out that it is also possible to construct local conserved quantities out of a linear superposition of the conserved mode occupation numbers, see e.g.
F. H. L. Essler and M. Fagotti, J. Stat. Mech. 064002 (2016).

R3: "As discussed in Ref. [41] in the non-equilibrium spectral function, a reasonable measurable quantity derived from the Green function by Fourier transform with respect to the relative time of the two fields, this "universal" decay is masked by other terms. These show typical Luttinger liquid power-law decay in t with interaction dependent exponents which turn out to be generically larger than -2. Surprisingly, this problem is not even mentioned in the present manuscript."
A3: We thank the referee for the useful suggestion. A discussion of the fermionic non-equilibrium spectral function can be very insightful for the reader. In Section 4 of the revised manuscript we now present and discuss this quantity in details.
In particular, we demonstrate that, at finite temperature, the non-equilibrium spectral function still shows universal power-law decay $\propto t^{-2}$. Moreover, this quantity exhibits marked differences with respect to the zero temperature case. Indeed, here the non-universal contributions are associated with a fast exponential decay towards the steady state value, whereas quench-induced features result always in a robust $\propto t^{-2}$ decay. A detailed analysis of the non-equilibrium spectral function is presented and it is also helpful to better clarify the importance of our new results and the role played by the initial finite temperature.

R4: "Besides this Eq. (30) contains the greater Green function of the isolated probe. In Eq. (31) the authors give an analytic expression for this. I am puzzled that via \omega_f and K_f this (non-interacting) Green function contains information about the interaction strength in the 1D system? In fact, in Eq. (A5) of Ref. [41], a publication which already contains the idea of the modified setup, the authors present an expression for the greater Green function of the isolated probe (in this case for T=0) which is independent of the interaction in the system. To me this appears to be more reasonable. Can the authors comment on this?"
A4: In the previous version of our paper we have expressed a non-interacting quantity, the probe Green function, in terms of parameters of the post-quench system Hamiltonian in order to introduce common energy and time scales. We want to stress that Eq.(30) is correct, only presented in a non-standard -- and perhaps misleading -- form.
To avoid any ambiguity, we have now re-written this expression in terms of the non-interacting probe parameters only.

R5: "After modifying the setup by including the probe a conceptual difficulty arises. With the coupling of the 1D interacting system to the infinite probe reservoir held in thermal equilibrium the authors no longer consider an isolated quantum system but rather an open one. (...) If I am mistaken the authors must provide an alternative way how to circumvent this conceptual problem of using closed system results in an open system setup. In any case I am very much surprised that the authors do not explicitly mention this type of conceptual difficulty."
A5: We definitely agree with the Referee that, if the system was coupled to infinite probe reservoir, it would eventually equilibrate to a thermal state with a relaxation dynamics strongly affected by the system-reservoir coupling. However, this is not the setup we have in mind and we have unintentionally created ambiguity by writing "kept at a fixed temperature T" when referring to the probe. Indeed, in our work we treat the channel and the probe on the same level, as two isolated systems initially prepared (for t<0) in a thermal ensemble at temperature T and T_p respectively. At t=0, the interaction is quenched in the channel and the two sub-parts of the setup are weakly tunnel-coupled. For t>0, channel and probe act as a closed system, essentially isolated from any external environment, and thus they evolve according to the post-quench Hamiltonian. Since we are interested in the relaxation dynamics of the channel, the probe is definitely non invasive and solely weakly coupled to it and the tunneling event can be safely described by exploiting a perturbative approach, neglecting all possible back-action effects. A possible candidate where such a setup can be envisioned is a system of two cold-atom channels, as this seems to naturally satisfy the requirements of tunability on one hand and strong de-coupling from external reservoirs.

R6: "The coupling to the probe in addition induces a local inhomogeneity to the Luttinger liquid which might affect the dynamics. It is well established that local inhomogeneities strongly change the equilibrium low-energy physics of Luttinger liquids. Can the authors exclude that this is an issue in the non-equilibrium dynamics of the suggested setup as well? Again the computational tool, namely perturbation theory in the system-reservoir coupling might be insufficient to capture and/or detect the proper impurity physics. The quench studied is not only one of the global interaction but at the same time a local single-particle parameter is changed (tunneling). Quenches of local parameters in the TL model (and related lattice models) were studied earlier. These studies might provide guidance for what to expect in the present case.”
A6: We agree with the Referee that local inhomogeneities, such as impurities, may strongly affect the properties of equilibrium Luttinger liquids (LLs). However, as stated in our previous answer, the probe is intended to be minimally invasive, in analogy, for instance, with a STM probe in a condensed matter setup. In the latter case, it has been shown by Aristov et al., PRL 105, 266404 (2010) that, for a point-like tunnel-coupling (such as the one considered in our work), effects of an external probe are far less dramatic than the ones induced by a local inhomogeneity. Moreover, to further support our results, we would like to point out that the problem of the quench of a weak single impurity in a homogeneous LL, for the zero temperature case, was discussed in Schiro & Mitra, PRB 91, 235126 (2015). Here, they showed that, regardless the quench amplitude, the RG flow associated with a potential backscattering term induced by the impurity is effectively cut by the energy scale set by the quench. They thus concluded that, differently from the equilibrium case, a weak impurity does not significantly disturbs a homogeneous quenched LL, even at zero-temperature.

R7: "Even if one ignores the above issues for the moment one might be tempted to conclude that the progress presented in the present manuscript (t^{-2} decay of the cross correlator for T>0) is rather small as compared to what (a subclass of) the authors already reported on in Ref. [41] (t^{-2} decay of the cross correlator for T=0). Can the authors make a stronger point why the extension of the T=0 result justifies another publication?"
A7: We do believe that our paper contains at least two important results. First of all, the persistence of the leading universal $ t^{-2} $ power law in the fermionic properties signals the survival of the quench-induced entanglement even in the case of a initial thermal preparation of the system. This alone is a remarkable result, especially in view of the fact that a quantum quench brings a strong memory of the pre-quench state (in this case, a thermal one) into the post-quench dynamics. It was therefore not trivial to expect the survival of the universal power law. Moreover, the preparation into a thermal state even helps the visibility of this power law: When the system is prepared in a thermal state the non-universal sub-leading power laws become exponentially suppressed. This is shown in more detail, in the revised version by looking at the fermionic non equilibrium spectral function and inspecting its long-time behavior.
This leaves ample room for the universal contributions to emerge and dominate the decay dynamics. In the revised version of our paper, we further elaborate on this point by analyzing in details what happens in a system composed by a channel and a probe at different temperatures. In particular, we argue that a higher initial temperature of the probe in comparison to the one of the system can enhance significantly the visibility of the universal power-law decay induced by the quench.
We conclude by noting that having established the validity of our previous results for systems prepared at non-zero temperatures could be also relevant in view of experimental realization and test of predictions based on the presence of quantum quench.
* * *
REPORT 2

R1: "The authors stress the fact that this contribution arises from the bosonic cross correlation term which indeed is related to the quench protocol. Honestly, I’m not much surprised by the fact that the cross correlation survives even for a thermal initial state, since it is merely a consequence of the quench protocol."
A1: It is true that the universal scaling is a pure consequence of the quench procedure. However, what is not so trivial is that entanglement between counter-propagating excitations survives in a detectable way even in the presence of a thermal preparation of the channel. In our opinion, this fact makes the result very interesting, especially in view of realistic experimental realizations. Moreover, although one could expect finite bosonic cross-correlators also at finite temperature in a quenched 1D system, universal quench-induced features in the fermionic channel are non-trivial at all at non-zero temperature. We have extensively clarified this point in the new revision.

R2: "Regarding this section (namely Sec. 3), I think it’s clearly written, nevertheless, I was struggling by figuring out how the (\tau/2t)^2 behavior in Eq.(24) for \tau << t << T^-1 comes from Eq. (22)."
A2: We have clarified the derivation of this result providing now more details in the manuscript.

R3: "Moreover, I would like to draw the attention of the authors to a strongly related result about the low-energy description of an interaction quench in the XXZ spin chain (Phys. Rev. B 92, 125131 (2015))."
A3: We thank the Referee for pointing out this reference. Indeed, also in the suggested paper deviations from the usual, Luttinger liquid-like power-law scaling of longitudinal correlation functions are observed, with an oscillatory behavior enveloped by an exponential decay. We now make connection to this result by quoting this paper.

R4: "The authors state that the fermionic field of the probe “is kept at fixed temperature”. Now I’m a bit confused: (1) is T the same temperature at which the original system has been prepared? What about different temperatures?"
A4: Indeed, this phrasing is misleading and we have revised the manuscript in order to remove any ambiguity about the setup considered. Since this criticism has been raised in a similar way also in Report 1, we kindly refer the Referee to the more detailed answer provided above (see R5-A5 in the replies to Report 1) and to the paper which now is much clearer about this point.

R5: "(2) When the authors claim that the probe is at thermal equilibrium, what do they exactly mean? In other words, I suppose the probe field is a new dynamical variable of the new setup, which evolves according to the new post-quench Hamiltonian. Is this the case?"
A5: Indeed, this is the case. As we have stressed in answering to Report 1, in the setup described in Sec. 4 of the previous version we consider the Luttinger liquid (LL) and the probe on the same level (note that in this revision we even consider the more general case of different initial temperatures for the channel and the probe). They are both treated as isolated systems prepared in a thermal state for t<0. For t>0, the whole system is composed of the interaction-quenched channel, tunnel-coupled to the non-interacting probe. As such, the probe evolves according to the post-quench Hamiltonian as well. Crucially, however, to the lowest perturbative order all transport properties are not affected by the back-action of the quenched channel on the probe degrees of freedom. The focus of our paper is the relaxation dynamics of the fermionic channel after a quantum quench, thus the probe is kept as non-invasive as possible and the lowest order perturbative approach is fully justified in this case. We briefly discuss this point in the revised manuscript.

R6: "Otherwise, if the probe is really kept at fixed temperature, then in the new setup, the system is no longer a closed system. Therefore, although there could be an intermediate regime for which the system relaxes toward a generalized thermal ensemble, at very large time, due to the influence of the external bath, I expect the system thermalizing. Maybe thermalization occurring starting from x_0, with a sort of light-cone effect. Can the authors be more clear about this.”
A6: As discussed in the previous point, the probe is not really kept at a fixed temperature and therefore no "conventional" thermalization is expected in the system. Related to this point, we believe that as long as the probe-channel coupling is weak - such as in the case of a non-invasive probe like the one we want to consider here - the GGE discussed in the text is a faithful description of the asymptotic regime of the system and that the universal power law is a robust phenomenon. In the revised manuscript we now discuss this issue.

R7: "In particular, I’m really curious about the effect of the new setup regarding the “local” quench with the probe. Indeed, as far as the global quench is joined with a local quench, I expect that, on top of the homogeneous dynamics induced by the global quench, there should be a sort of spreading of particles density injected in x_0. This leading to two different stationary descriptions, inside and outside the light cone."
A7: We do agree with the Referee: on top of the global effect due to the homogeneous quench of interactions in the channel, a light-cone effect originates from the position where channel and probe are tunnel-coupled. In this respect, in a non-quenched system some of the Authors have considered the time-resolved fractionalization of injected particles in a helical LL - see
A. Calzona, M. Carrega, G. Dolcetto, and M. Sassetti Phys. Rev. B 92, 195414 (2015)
which leads to two charge and spin packets counter-propagating through the system. We expect that this behavior survives even in the presence of a homogeneous interaction quench, thus creating the light-cone physics also expected by the Referee.

---

## Round 2 · List of Changes

• The abstract has been modified, in order to reflect all the changes in the main text and to highlight the new results presented in the resubmitted version of the paper (non-equilibrium spectral function and different temperatures for probe and 1D channel).

  • The final part of the introduction has been partly re-written to better describe the new results and the main task of our work. In particular, the results concerning the non-equilibrium spectral function and the case of different probe and channel temperatures now addressed in the manuscript, are recapitulated here.

  • In Section 2 ("Model") we now mention, after Eq. (3), the cut-off procedure employed in the paper and we make reference to other possible approaches (constructive bosonization), following the suggestions of Report 1.

  • In Section 2 ("Model"), the new Eq. (11) is provided which replaces former Eqs. (10-11) and contains, in combination with Eq. (3), the same amount of information.

  • Section 3 ("Fermionic and bosonic correlation functions") has been updated. We now provide more details about the derivation of our results, with new Eqs. (12,13,15-18) allowing for a complete discussion of all regimes of the Green's function we consider. In addition, new Eqs. (28,30) allow a complete discussion of the dynamics of the bosonic cross-correlators.

  • A new Section 4 ("Non-equilibrium spectral function") has been added. This is devoted to a study of the (fermionic) non-equilibrium spectral function, as suggested in Report 1. It contains new results not discussed previously elsewhere. We define and discuss the local lesser non-equilibrium spectral function, which requires also the new material presented in Section 2. The steady state of this quantity, and its asymptotic behavior as a function of time, are analyzed in details. New Eqs. (31-45) have been added, as well as new Fig. (2).

  • New Section 5 ("Transport properties") replaces old Section 4 (same name). However, we now extend our results to the more general case of a system where the probe and the channel are prepared (for t<0) at different temperatures. Several equations have been updated to reflect this generalization and the discussion has been expanded in order to describe the new physics. In particular, new Eqs. (59-61) and new Fig. 5 are provided in order to discuss also the case of different probe and channel temperatures.

  • We have added new Refs.: 23, 34, 42, 43, 52, 53, 57, 58.

---

## Round 3 · Author Response

We would like to thank both Referees for their overall positive assessment of our work.

In this new version, following the referee suggestions, we have made small amendments and corrected some typos, as detailed below. In the following, we reply in details to the criticisms raised in "Anonymous Report 2 on 2018-3-28". With best regards,

Alessio Calzona on behalf of all the Authors.

Anonymous Report 2 on 2018-3-28

R: "To enhance the relevance of the work which was questioned by both referees they added a chapter on the absolute time-dependence of the local "spectral function" (Fourier transform of the local Green function with respect to the relative time). This indeed provides further insights. However, one might argue that the chapter on the transport setup (see 2.) is now superfluous. With respect to this new chapter I am puzzled by the first line of Eq. (35). While on the left hand side the limit t -> oo was taken, t appears explicitly on the right hand side!?"

A: We thank the Referee for pointing out this typo. Indeed, Eq. (35) describes the long time limit behavior of the non-equilibrium spectral function. In the revised manuscript we have added the limit t->infty also in the right hand side of Eq. (35). Concerning the chapter on the transport setup, we strongly believe that it contains important information and results on its own. Indeed, although the universal power-law decay emerges already in the spectral properties of the system, transport properties represent a convenient way to actually probe the universal behavior.

R: "2. The authors made an effort to now explain more clearly the transport setup. Both referees were confused by the original description. However, I am still not fully convinced that the field theory the authors study can be realized in any microscopic model (not to speak of experiments). In PRL 105, 266404 mentioned in the authors reply it is emphasized that "...we have shown that the picture of tunneling into a LL is qualitatively modified when the tunneling amplitude is not treated as infinitesimally small. The conventional FP [fixed point] has a finite basin of attraction only in the model of the point tunnel contact, but taking a finite size of the contact (or any perturbation induced by the contact in the wire) into account makes it unstable." If the "conventional FP" is generically unstable (as stated in this paper) the probe will generically be invasive and the physics on asymptotic time scales will not be the one of the unperturbed system (the probe does not simply probe the system). In any case the authors revisions are in this respect inappropriate: "It is worth to underline that the probe is a tool to inspect the intrinsic properties of the fermionic channel out-of-equilibrium and, thus, it is supposed to be as non-invasive as possible. A perturbative approach in the weak tunnel coupling is therefore fully justified in evaluating the related transport properties." This paragraph does not contain any arguments that a proper "non-invasive" limit exists at all. The latter is the question of utter relevance (see my quote from the paper brought up by the authors). At the minimum the authors must properly revise the wording and backup their view by the proper references. "

A: We do certainly agree with the Referee when he quotes the paper by Aristov et al. [PRL 105, 266404 (2010)]. However, we would like to emphasize that the fact that the conventional fixes point (FP) is generically unstable is not in contradiction with the existence of a "non-invasive probe limit". Indeed, in real experiments the presence of both a finite temperature and a finite bias fix an energy scale at which the renormalization group (RG) flow is cut off. Therefore, the (stable) FP associated with the breakup of the wire into two independent semi-infinite wires is, in general, never reached by the RG flow. That is what actually makes it possible to perform experiments such as the one in Ref. 62 [Nano Letters 25, 3684 (2015)], where an STM tip is used to probe a quantum wire without breaking it into two uncoupled pieces. Moreover, as discussed in PRB 91, 235126 (2015), we know that the quench introduces a further energy scale which stops the flow to the breakup FP. All this means that a set of parameters (temperature, bias, tunnel-coupling, quench) in which the presence of the probe is non-invasive must exist. To clarify this point we have added a note in our manuscript describing in more details what we mean by "non-invasive" probe.

---

## Round 3 · List of Changes

- we have added $lim_{t\to\infty}$ in the right hand side of Eq. (35)
- we have added a note at the beginning of Section 5 which better clarifies the concept of "non-invasive" probe.
- we have added Ref. 42, 60 and 62.

---

## Editorial Decision

published